# Novel Fluorescent Strategy for Discriminating T and B Lymphocytes Using Transport System

**DOI:** 10.3390/pharmaceutics16030424

**Published:** 2024-03-19

**Authors:** Heewon Cho, Na-Kyeong Hong, Young-Tae Chang

**Affiliations:** Department of Chemistry, Pohang University of Science and Technology (POSTECH), Pohang 37673, Gyeongsangbuk-do, Republic of Korea; heewon@postech.ac.kr (H.C.); nkhong5868@postech.ac.kr (N.-K.H.)

**Keywords:** fluorescent carbohydrate library, high-throughput phenotypic screening, gating-oriented live-cell distinction, solute carrier, B cell-specific probe

## Abstract

Fluorescent bioprobes are invaluable tools for visualizing live cells and deciphering complex biological processes by targeting intracellular biomarkers without disrupting cellular functions. In addition to protein-binding concepts, fluorescent probes utilize various mechanisms, including membrane, metabolism, and gating-oriented strategies. This study introduces a novel fluorescent mechanism distinct from existing ways. Here, we developed a B cell selective probe, CDrB, with unique transport mechanisms. Through SLC-CRISPRa screening, we identified two transporters, SLCO1B3 and SLC25A41, by sorting out populations exhibiting higher and lower fluorescence intensities, respectively, demonstrating contrasting activities. We confirmed that SLCO1B3, with comparable expression levels in T and B cells, facilitates the transport of CDrB into cells, while SLC25A41, overexpressed in T lymphocytes, actively exports CDrB. This observation suggests that SLC25A41 plays a crucial role in discriminating between T and B lymphocytes. Furthermore, it reveals the potential for the reversible localization of SLC25A41 to demonstrate its distinct activity. This study is the first report to unveil a novel strategy of SLC by exporting the probe. We anticipate that this research will open up new avenues for developing fluorescent probes.

## 1. Introduction

Discriminating between cell types is a crucial step in understanding complex biological events. While antibodies are commonly used to differentiate cell types, their recognition is limited to surface markers. In contrast, fluorescent small molecules can easily penetrate cells without causing significant disruption and can detect intracellular biomarkers [1,2,3,4,5]. Motivated by the identification of a vast array of potential targets using fluorescent chemicals, our group has established Diversity-Oriented Fluorescence Library (DOFL) to extend the scope of detection. Through this initiative, we have made substantial contributions to the field, having developed over 30 bioprobes (Figure 1) [6]. Conventionally, fluorescent probe development focuses on binding to proteins, a method we termed Protein-Oriented Live-cell Distinction (POLD) [7,8,9,10]. Throughout the procedure, the protein-binding system encounters limitations in explaining the journey of fluorescent molecules, particularly in the absence of protein partners. As an alternative approach, we have shifted and expanded our perspectives to a novel strategy [11]. We realized that some probes can target molecules outside of cells by staining carbohydrates (Carbohydrate-Oriented Live-cell Distinction: COLD) [12] or membranes (Lipid-Oriented Live-cell Distinction: LOLD) [13]. The representative example of LOLD is CDgB [14], exhibiting selectivity to B over T cells based on membrane flexibility. Furthermore, we found that other probes can serve as substrates for specific enzymes, acting as turn-on sensors. This mechanism, termed Metabolism-Oriented Live-cell Distinction (MOLD) [15], offers different perspectives. In addition to these mechanisms, we have explored a distinct strategy involving transporters that do not require specific partners inside the cells. Considering the wide range of substrates transferred by transporters, it would be plausible to consider them as potential gating partners for selective probes [16]. Transporters are broadly classified into solute carrier (SLC) and ATP-binding cassette (ABC) families. SLC transporters primarily facilitate the uptake of small molecules into cells (pump in) [17], while ABC transporters are responsible for exporting substrates to the extracellular space (pump out) [18]. Leveraging transporter-dependent mechanisms, we have developed several selective bioprobes [11]; the mechanisms reliant on the transporters uncovered have not fully encompassed all transporter pools yet. 

This article presents a newly elucidated fluorescent strategy that stands apart from the existing concepts. To identify this novel mechanism, we first screened compounds in murine spleen, leading to the elicitation of CDrB, which effectively distinguished B from T cells. Through SLC-CRISPRa screening, we identified a unique SLC transporter, SLC25A41 that can pump out CDrB, along with SLCO1B3, transporting CDrB into cells. In addition, we validated that SLC25A41 could be a biomarker for T lymphocytes compared to SLCO1B3, which is pervasive in both T and B cells. Moreover, we newly propose that SLC25A41 could be reversibly located to properly expel its substrate. 

## 2. Materials and Methods

### 2.1. Animal Experiment

Our research complies with all relevant ethical guidelines. All animal experimental protocols were performed in compliance with the Guidelines of the Pohang University of Science and Technology (POSTECH) Animal Care and Use committee (Approval No. POSTECH-2023-0060). All the mice were maintained in the animal facility of the Pohang University of Science and Technology (POSTECH) Biotech Center in accordance with the Institutional Animal Care and Use Committee of POSTECH. All the animal experiments were performed according to the recommended guidelines.

### 2.2. Lymphocyte Preparation 

Six-to-eight-week-old male wild-type C57BL/6 mice were purchased from Pohang University of Science and Technology. To acquire the murine lymphocytes, tissues were first harvested and homogenized. Then, red blood cells were removed using RBC lysis buffer (Thermo Fisher Scientific, Rockford, IL, USA) and centrifuged for 5 min at 1500 rpm. After washing the sample, the cell pellet was resuspended in RPMI1640 Medium (with 2.5 g/mL glucose, Gibco, Waltham, MA, USA) containing 10% Heat-Inactivated Fetal Bovine Serum (Gibco) and 1% Penicillin Streptomycin (WELGENE, Gyeongsan-si, Republic of Korea). 

### 2.3. Flow Cytometry-Based Screening

The splenocytes were resuspended in the RPMI1640 Medium (with 2.5 g/mL glucose, Gibco) containing 10% Heat-Inactivated Fetal Bovine Serum (Gibco) and 1% Penicillin Streptomycin (WELGENE). Then, the splenocytes were seeded into 5 mL tubes (2 × 10^5^/tube) and incubated with library compounds at a concentration of 1 μM. After 1 h, the samples were read using a LSR-II cytometer (BD, Franklin Lakes, NJ, USA). The gating strategies were employed as described below. Firstly, a FSC-A versus SSC-A plot was used to gate and analyze lymphocytes. Singlets were picked up by FSC-A versus FSC-H, and fluorescent channels were applied. A data analysis was performed using flowJo software (10.7.1 version, BD, Franklin Lakes, NJ, USA). To screen the LC library, we used five fluorescence channels: BUV395 (348 nm/395 nm), FITC (494 nm/519 nm), PE (496 nm/578 nm), APC (650 nm/660 nm), and AF700 (696 nm/719 nm). During the analysis, the gating strategies were used as described below. 



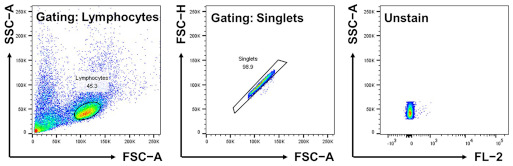



### 2.4. Isolation of B Cells

The single cells were collected from the spleen, and they were lysed using RBC lysis buffer (Thermo Fisher Scientific, Rockford, IL, USA). The collected cells were then incubated with biotinylated monoclonal antibodies for 15 min at 4 °C. To isolate B cells, a Mouse B Lymphocyte Set-DM (BD Bioscience Co., Franklin Lakes, NJ, USA) was used. The cells were washed with 1X BD IMag™ Buffer (10× buffer was diluted with deionized water, BD Bioscience Co., Franklin Lakes, NJ, USA) and centrifuged (1500 rpm for 5 min). Then, BD IMag™ Streptavidin Particle Plus—DM (BD Bioscience Co., Franklin Lakes, NJ, USA) was added to cells bearing biotinylated antibodies. After 30 min, the tube containing the labelled cell suspension was placed within the magnetic field of a BD IMagnet™ (BD Bioscience Co., Franklin Lakes, NJ, USA) with IMag buffer. After 7 min, antibody unlabeled cells were collected, and this process was repeated three times. For further research, collected cells were centrifuged (1500 rpm for 5 min) and resuspended with cell media. 

### 2.5. Isolation of T Cells

The single cells were collected from the spleen, and they were lysed using RBC lysis buffer (Thermo Fisher Scientific, Rockford, IL, USA). The collected cells were then incubated with biotinylated monoclonal antibodies for 15 min at 4 °C. To isolate T cells, a Mouse T Lymphocyte Set-DM (BD Bioscience Co., Franklin Lakes, NJ, USA) was used. The cells were washed with 1X BD IMag™ Buffer (10× buffer was diluted with deionized water, BD Bioscience Co., Franklin Lakes, NJ, USA) and centrifuged (1500 rpm for 5 min). Then, BD IMag™ Streptavidin Particle Plus—DM (BD Bioscience Co., Franklin Lakes, NJ, USA) was added to cells bearing biotinylated antibodies. After 30 min, the tube containing the labelled cell suspension was placed within the magnetic field of a BD IMagnet™ (BD Bioscience Co., Franklin Lakes, NJ, USA) with IMag buffer. After 7 min, antibody unlabeled cells were collected, and this process was repeated three times. For further research, collected cells were centrifuged (1500 rpm for 5 min) and resuspended with cell media.

### 2.6. Generation of SLC-CRISPRa Pools

CRISPR-SLCa pools were generated based on the previous method. HeLa cells (ATCC^®^ CCL-2TM) were transfected with dCas9-VPR and purified using G418 antibiotic (Invitrogen, Waltham, MA, USA, 500 μg/mL). Then, dCas9-VPR HeLa cells were infected using lentiviral library plasmids to stably overexpress the 3800 sgRNA libraries, generating the SLC-CRISPRa pools. Then, puromycin (2 μg/mL) was added to purify the transfected cells. To keep the heterogeneity of the SLC-CRISPRa pools, more than 1.5 × 10^6^ cells were seeded for further subculture.

### 2.7. Cell Sorting 

SLC-CRISPRapools were incubated with CDrB (0.5 μM) for 30 min. Then, Trypsin-EDTA (Welgene) was used when detaching the CRISPR-SLCa pools. For the screening, gating strategies were employed as described below. Firstly, cells were selected via FSC-A versus SSC-A. Then, FSC-A versus FSC-H was used to detect singlets, and a fluorescent channel (FL-2: 561 nm/578 nm) was applied. The live singlets of the cell population showing the 3% brighter/dimmer populations in CRISPR-SLCa libraries, respectively, were sorted out using a S3e cell sorter (Bio-Rad, Hercules, CA, USA). The sorted cells were expanded by culturing for the next round screening until the enrichment reached around 95%.



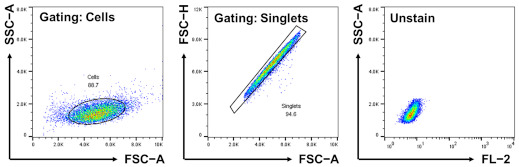



### 2.8. RT-PCR 

The total RNA was extracted using an RNeasy Mini Kit (QIAGEN Inc., Germantown, MD, USA) and the amount and quality were measured using a Nanodrop 2000 (Thermo Scientific, Waltham, MA, USA). cDNA was synthesized with a High Capacity cDNA Reverse Transcription Kit (Applied Biosystems, Waltham, MA, USA) according to the instructions. A qRT-PCR was carried out using a TB Green™ Premix Ex Taq™ II (Tli RNaseH Plus) Kit (TaKaRa). The reactions were run on a qTOWER3 Real-time PCR system (Analytic Jena, Jena, Germany) with the following cycles: 10 min at 95 °C, and 40 cycles of 15 s at 95 °C and 1 min at 55 °C. The experiment was repeated three times individually. No data were excluded from the analyses. The data were collected through qPCRsoft 4.0 (Analytic Jena). The designed primers were as follows: human slc25a41, F-5′-CTGGAAGTGGATAACAAGGAGGC-3′ and R-5′-GGTGAAGTTCGTCTTGGAGGAG-3′; human slco1b3, F-5′-GGATGGACTTGTTGCAGTTG-3′ and R-5′-TTAGTTGGCAGCAGCATTGT-3′; human β-actin, F-5′-GGATGCAGAAGGAGATCACTG-3′ and R-5′-CGATCCACACGGAGTACTTG-3′; mouse slc25a41, F-5′-TGACTCTACGCAGAACTGGC-3′ and R-5′-GACCTTCATGAAGTTGGGGG-3′; mouse slco1b2, F-5′-TGGAAGGCATAGGGTAGGCGGT-3′ and R-5′-TGGGCAGCTTTGCTTGGATGCT-3′; and mouse GAPDH, F-5′- TGTCCGTCGTGGATCTGAC-3′ and R-5′- CCTGCTTCACCACCTTCTTG-3′.

### 2.9. Chemical Materials and General Methods for CDrB Synthesis

All used compounds and solvents were purchased from Alfa Aesar (Haverhill, MA, USA), Sigma Aldrich (St. Louis, MO, USA), Combi-Blocks (San Diego, CA, USA), TCI (Tokyo, Japan), or Samchun Chemicals (Seoul, Republic of Korea). All the chemicals were directly used without further purification. MERCK silica gel 60 (230–400 mesh, 0.040–0.063 mm) was used for normal-phase column chromatography. The optical properties were performed with a SpectraMax M2e spectrophotometer (Molecular Devices, Silicon Valley, CA, USA ) in a 96-well plate (clear bottom) and QS high-precision cuvette. The relative fluorescence quantum yield method was selected, and sulforhodamine 101 (Φ = 0.9) was utilized as the standard. The quantum yield equation was calculated using Equation (1). For an analytical characterization of CDrB HPLC (Agilnet, Santa Clara, CA, USA, 1260 series), a DAD (diode array detector) and a single quadrupole mass spectrometer (Aglient, 6100 series, ESI) were used. Eluents (A: H_2_O with 0.1% formic acid (FA), B: MeCN with 0.1% FA) and a Zorbax SB-C18 column (2.1 × 50 mm, 1.8 μm particle size, 80 Å pore size) were used. High-performance liquid chromatography (HPLC) was utilized on Prep. HPLC (Shimadzu, Kyoto, Japan) with a PDA detector with a C18(2) Luna column (5 μm, 250 mm × 21.2 mm, 100 Å). A gradient elution of 20% B to 65% B for 15 min and then 65% B to 99% B for 52 min was used at a flow rate of 15 mL/min (solvent A: H_2_O; B: MeOH). 1H and 13C NMR spectra were obtained from Brucker AVANCE III HD 850.
Φ_fi_ =(*F*_i_/*F*_s_)(*f*_s_/*f*_i_)(n_i_/n_s_)^2^•Φ_f_^s^(1)
where Φ_fi_ and Φ_fs_ represent the fluorescence quantum yield of the sample and standard, respectively. *F* represents the area under the curve of the fluorescence spectrum (from 550 to 800 nm), n represents the refractive index of the solvent, and f represents the absorption factor (*f* = 1 − 10^–A^, where A represents the absorbance) at the excitation wavelength selected for the sample and standard.

### 2.10. Synthesis of CDrB

The reaction followed the following procedure. 6-amino-6-deoxy-D-glucose (36.8 mg, 205.2 μmole), BODIPY fluorophore (30 mg, 64.8 μmole), HATU (78 mg, 205.2 μmole), and DIEA (59.5 μL, 342 μmole) were dissolved in DMF (20 mL) and stirred for 2 h, and the mixture was acidified with 1 M HCl and extracted three times with DCM. The collected mixture was washed twice with brine, dried over Na_2_SO_4_, filtered, and concentrated in vacuo. The residue was purified via silica gel chromatography using a gradient of MeOH:DCM=1:20 to 1:5. Then, high-performance liquid chromatography (HPLC) was utilized on Prep. HPLC (Shimadzu) with a PDA detector with a C18(2) Luna column (5 μm, 250 mm × 21.2 mm, 100 Å). A gradient elution of 20% B to 65% B for 15 min and then 65% B to 99% B for 52 min was used at a flow rate of 15 mL/min (solvent A: H_2_O; B: MeOH). After purification, a purple solid was obtained (11.6 mg, 30%). 1H NMR (850 MHz, Methanol-d4) δ 7.55 (d, J = 8.1 Hz, 2H), 7.46 (d, J = 16.4 Hz, 2H), 7.34 (d, J = 16.4 Hz, 2H), 6.97 (d, J = 8.5 Hz, 2H), 6.85 (s, 1H), 6.18 (s, 1H), 5.11 (d, J = 2.89 Hz, H_1α_), 4.65 (d, 18.4 Hz, H_2α_), 4.62 (s, H_1β_), 4.15 (s, H_6β_), 4.01 (s, H_5β_), 3.94 (s, H_6α_), 3.85 (s, 3H), 3.81 (s, H_2β_), 3.76 (s, H_5α_), 3.73 (s, H_3β_), 3.70 (d, J = 11 Hz, H_4α_), 3.63 (s, H_4β_), 3.59 (s, H3_α_), 2.59-2.50 (m, 13H). ^13^C NMR (214 MHz, Methanol-*d_4_*) δ 173.57, 172.27, 160.72, 153.17, 152.65, 143.02, 140.98, 140.19, 135.85, 135.46, 131.24, 129.37, 128.44, 121.23, 117.91, 116.31, 114.04, 101.84, 75.19, 73.61, 72.58, 70.63, 66.47, 62.82, 62.02, 61.26, 54.44, 37.60, 36.40, 23.47, 22.83, 15.62, 15.55, 15.28. LC-MS (ESI) [M+Na]^+^, m/z calcd for C_30_H_36_BF_2_N_3_NaO_7_ 622.2, found: 622.2. 



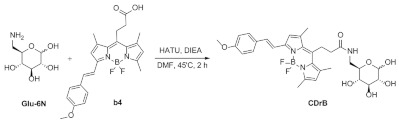



## 3. Results

### 3.1. Eliciting B Cell Selective Probe

We established a complex screening system using murine spleen to elicit a probe, designed it for the single cells of splenocytes, and incubated them with the Luminescent Carbohydrate (LC) library, hypothesizing that the main energy sources of carbohydrates may show distinctive selectivity along with fluorophore effects [19]. After 1 h, the stained cells were read via flow cytometry, and extracted numeric values from graphs were calculated using the stain index equation (Figure 2A). The analyzed data were plotted in a one-dimensional graph, and showed one compound that had the highest stain index among 80 compounds (Figure 2B,C). This molecule consisted of a glucose conjugated with a red fluorophore at 6-position, dubbed as CDrB (Compound Designation red B) (Figure 2D). To confirm the selectivity, splenocytes, composed of B cells (~70%) and T lymphocytes (~30%) [20], were co-stained with CDrB after adding anti-CD3, a T cell antibody (Figure 2E). The result revealed that CDrB strongly stained B lymphocytes possessing a higher stain index of 2.8 compared to previously reported probes, CDgB (SI:1.25) [14]. To scrutinize the selectivity of CDrB independent of an environment, T and B lymphocytes were isolated via magnetic-activated cell sorting from murine spleens (Figure 2F). Then, each cell population was incubated with CDrB for 1 h. It showed that CDrB brightly visualized B lymphocytes three times more than T cells. 

### 3.2. CDrB Selective Mechanism Identification

Inspired by the selectivity, we carried out the SLC-CRISPRa system, overexpressing each individual SLC transporter. This library consisted of 10 single-guide RNAs (sgRNAs) corresponding to 380 transporters, resulting in a total of 3800 variants [21]. After staining the library with CDrB, we iteratively sorted out 3–5% of the brighter populations until enrichment (Appendix A). Moreover, we picked out 3–5% of dimmer cell groups in a novel attempt (Appendix A), hypothesizing that they might offer new insights into SLC transporters. Clear shifts were observed compared to the unsorted pools in both sorting methodologies. Then, DNA sequences were analyzed using next generation sequencing (NGS). Surprisingly, we found enriched SLC targets, SLCO1B3 (99.9%) from the brightness sorting and SLC25A41 (94.3%) from the dimmer enrichment (Figure 3A). 

### 3.3. CDrB Selective Mechanism Validation

To verify if these transporters are indeed gating targets of CDrB, single-cell cloning was conducted. After the process, clones exhibiting the highest expression levels of the target transporters were selected (Appendix A). Then, CDrB-stained slco1b3- and slc25a41-overexpressing clones confirmed that the results were consistent with the CRISPR screening data. The brightest signal was observed in slco1b3-cloned cells, while the slc25a41-overexpressing clone showed the dimmest intensity of CDrB compared to the control cells (Figure 3B–D). We further investigated whether these results could be extrapolated to the actual model of T and B lymphocytes. We firstly isolated T and B lymphocytes via magnetic-activated cell sorting (MACS) and assessed the expression levels of target transporters. Interestingly, the slco1b3 expression did not display significant differences between B and T cells (Figure 3E), whereas slc25a41 [22,23], a mitochondrial carrier, was predominantly expressed in T over B lymphocytes (Figure 3F). This result suggests that SLC25A41 could be a potential gate for CDrB to differentiate T and B lymphocytes. To ensure clarity, we confirmed that CDrB localized in mitochondria in control cells (SLC-CRISPRa pools). This outcome implies that CDrB firstly transports through SLCO1B3, a plasma membrane transporter [24,25], and ultimately reaches mitochondria, but CDrB can be exported when encountering SLC25A41-overexpressing cells, such as T lymphocytes.

### 3.4. Proposed Mechanism of CDrB

Through the systematic screening of SLC-CRISPRa using two different approaches, we determined that CDrB could serve as a substrate for both SLCO1B3 and SLC25A41. In the case of SLCO1B3-overexpressing cells, CDrB was observed to accumulate within the cells (B cells), indicating that SLCO1B3 functions as an influx transporter with an inward location. However, cells highly expressing SLC25A41 (T cells), even those with SLCO1B3, showed a weak signal of CDrB, suggesting the export of CDrB with reversible localization, again termed as CDrB: Compound Designation reversible B (Figure 4). This mechanism represents a unique capability of SLC transporters to actively pump out their substrates with a reversible location. Moreover, CDrB was identified as a clear substrate that pumps out from SLC25A41, opening the door to an in-depth exploration of a Gating-Oriented Live-cell Distinction strategy.

## 4. Discussion

Our body comprises diverse sets of immune cells strategically playing their own roles in detecting foreign materials, such as bacteria, and removing them to maintain homeostasis. Among these, lymphocytes, including T and B lymphocytes, are core components of our immune system, as they have the ability to form immune memory and control immune reactions [26,27]. While T cells are mainly involved in cell-mediated immunity by regulating immune responses [28], B lymphocytes are primarily responsible for humoral immunity through antibody production [29]. Despite their importance, discriminating between T and B cells has been challenging due to their similar appearance. Although antibodies are commonly used, they are not suitable for monitoring cells in a live state and lack the ability to discover novel biomarkers, necessitating a new avenue to uncover the functions of cells. To overcome these limitations, our group focused on fluorescent small molecules to achieve the dual goals of detection and visual representation. Over decades, we have developed more than 30 fluorescent probes and identified diverse staining mechanisms, including binding targets, transporters, and carbohydrates [11,16]. Among these, we believe that the solute carrier family, regarding influx transporters, can offer new insights for discriminating T and B lymphocytes, as SLCs include more than 400 members and have a broad substrate specificity [30].

To ascertain adequacy, we selected murine splenocytes as the screening format, primarily composed of T and B cells [20], and finally elicited the B cell-selective probe, CDrB, by applying the LC library. CDrB exhibited the most distinct separation (stain index: 2.8) between T and B lymphocytes compared to previously reported B cell fluorescent molecules within 1 h. Inspired by the selectivity, we introduced CDrB into the SLC-CRISPRa library and approached it in two different ways to sort out brighter and dimmer populations, respectively. Surprisingly, we obtained two different target transporters: SLCO1B3 from the brightness sorting method and SLC25A41 from the dimmer enriched group. To clarify the staining mechanism of CDrB, we carried out single-cell cloning and successfully obtained SLCO1B3- and SLC25A41-overexpressing cells. While SLCO1B3-cloned cells, showing similar expression patterns in both T and B lymphocytes, displayed a strong signal of CDrB, SLC25A41-overexpressing cells including T lymphocytes exhibited a weak intensity. These results imply that SLCO1B3 and SLC25A41 have opposite functions regarding their substrates, further suggesting that the activation site of SLC25A41 reverses compared to that of SLCO1B3. This novel strategy of fluorescent probes for identifying a specific cell type in a complex system provides deeper insights into the physiological functions and interactions of cells. Considering that CDrB can discriminate between the two main adaptive immune cells, it may serve as a platform for drug development and disease treatment. Additionally, it can act as an early indicator of diseases, aiding in disease prevention and early diagnosis.

## 5. Conclusions

In this article, we presented a novel strategy of a B cell-selective probe, CDrB. Using systematic SLC-CRISPRa screening, we identified two transporters, SLCO1B3 and SLC25A41, employing distinct approaches. SLCO1B3-overexpressing cells exhibited the brightest CDrB signal, while the intensity was weakest in SLC25A41-cloned cells. These findings suggest that SLCO1B3 facilitates CDrB transport into cells, whereas SLC25A41 actively exports CDrB. The contrasting activities suggest that SLC25A41 may reversibly locate to effectively pump out CDrB. This study represents the first report on an SLC transporter expelling a fluorescent probe, with potential implications as a biomarker of T lymphocytes. We believe that this discovery provides new perspectives into the development of fluorescent probes. 

## Figures and Tables

**Figure 1 pharmaceutics-16-00424-f001:**
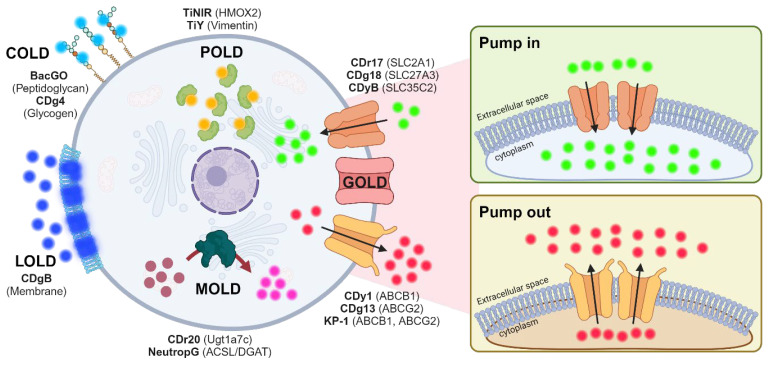
The selective staining mechanism of fluorescent probes. Fluorescent probes broadly have five strategies. Protein-Oriented Live-cell Distinction (POLD): probes have binding targets, especially proteins, inside cells; Carbohydrate-Oriented Live-cell Distinction (COLD): probes have selectivity to carbohydrates to discriminate specific cell types; Lipid-Oriented Live-cell Distinction (LOLD): compositions of phospholipids and cholesterol located in plasma membrane could be a guide to provide selectivity to probes; Metabolism-Oriented Live-cell Distinction (MOLD): metabolic activities can provide selectivity to probes to be trapped inside specific cell types; Gating-Oriented Live-cell Distinction (GOLD): transporters act as main gates for uptake (SLC; solute carrier family) or export (ABC; ATP-binding cassette) probes. Different expression levels of transporters can provide clues for solving the staining mechanisms of probes. This figure was created with BioRender.com.

**Figure 2 pharmaceutics-16-00424-f002:**
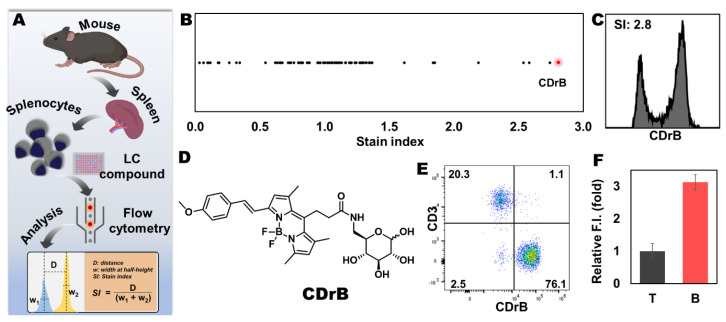
Development of B cell−selective probe. (**A**) A schematic view of flow cytometry-based screening. The spleen was firstly dissociated into single cells of splenocytes, and the cells were stained with LC (Luminescent Carbohydrate) compounds. After 1 h, the samples were read via flow cytometry, and extracted values were analyzed using the equation for calculating the stain index. (**B**) The calculated stain index, which shows separation grades in the splenocytes based on the intensity of LC compounds, was plotted in a one-dimensional graph. (**C**,**D**) The CDrB structure is displayed. (**E**) T cell antibody (CD3) shows a negative correlation with the CDrB signal. (**F**) The isolated T and B lymphocytes from murine spleen incubated with CDrB, and B cell selectivity is shown. Figure 2A was created with BioRender.com.

**Figure 3 pharmaceutics-16-00424-f003:**
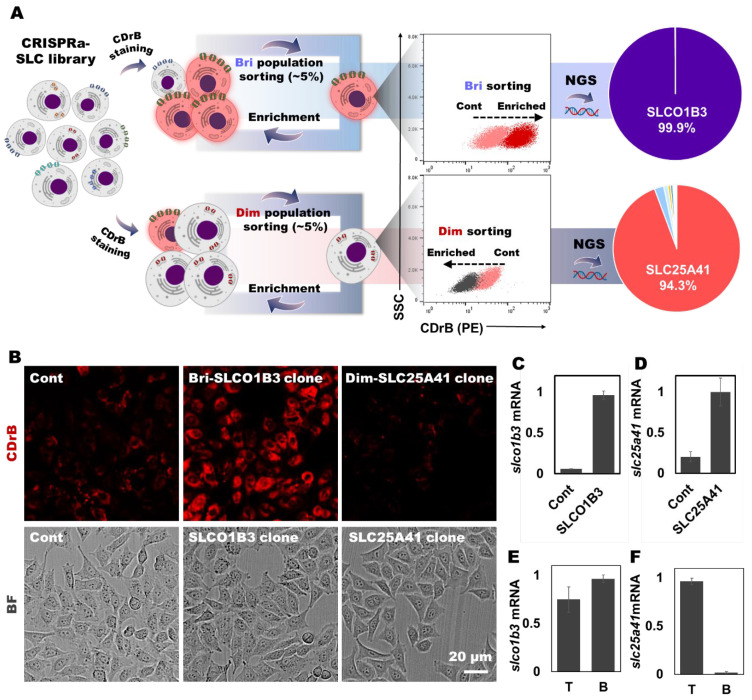
Transporter identification and validation through CRISPRa−SLC screening. (**A**) The schematic process of CRISPRa-SLC screening. CRISPRa-SLC library was stained with CDrB, and both sides of the ~3% brighter and dimmer populations were continuously sorted until enrichment. The enriched cells were analyzed via next generation sequencing (NGS), and the results are displayed as circular graphs. (**B**) With enriched populations, SLCO1B3- and SLC25A41-cloned cells were acquired compared to control cells. Three groups (the control and SLCO1B3- and SLC25A41-clone groups) were stained by CDrB. Then, images were acquired with a ×40 objective water lens. For these groups, the mRNA expression levels of the target transporters in the cloned cells, (**C**) slco1b3 and (**D**) slc25a41, were confirmed. To identify the trends of the expression levels of candidate transporters, murine T and B lymphocytes were isolated from the spleen using magnetic-activated cell sorting (MACS). The analysis results are displayed in (**E**) slco1b3 and (**F**) slc25a41. Figure 3A was created with BioRender.com.

**Figure 4 pharmaceutics-16-00424-f004:**
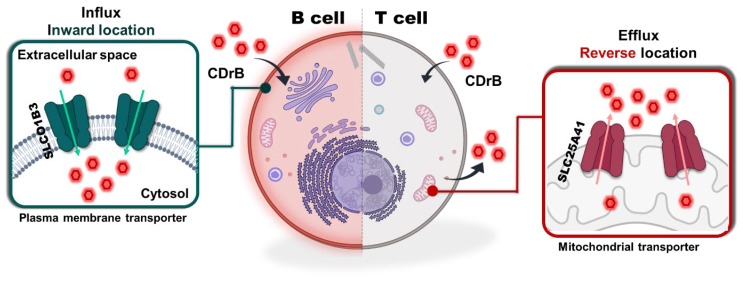
Proposed mechanism of CDrB. CDrB is taken up by SLCO1B3 within the cells, expresses in both T and B lymphocytes, and localizes within the mitochondria. However, T cells, which overexpress SLC25A41 compared to B lymphocytes, actively efflux CDrB. Based on this phenomenon, we hypothesize that SLC25A41 exhibits reversible localization to properly expel its substrate, in contrast to the inward positioning of the influx transporter SLCO1B3.

## Data Availability

The data presented in this study are available from the first author (H.C.) upon request.

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
