# Peer review of "Novel Fluorescent Strategy for Discriminating T and B Lymphocytes Using Transport System"

_pharmaceutics, 2024, doi:10.3390/pharmaceutics16030424_

Round 1
Reviewer 1 Report
Comments and Suggestions for Authors
The manuscript by Cho et al reports the use of a novel fluorescent probe that is transported by two different transporters of the SLC family and allows the flow cytometric discrimination between murine T- and B-cells. This is a repliminary work with potential interest for the use of live-cell probes for cytometric studies. However, the manuscript has important limitations.
The title is misleading and should be written more carefully. Actually, CDrB is not a B-cell selective probe, as it is taken up by SLCO1B3 which is present in both T- and B-cells. What makes the “labeling” selective of B cells is that T cells pump CDrB out due to overexpression of SLC25A41 mitochondrial transporter. Title should be more along the lines of selective identification of B cells based on the transport of CDrB probe.
The assumption that an alternative location (plasma membrane?) of SLC25A41 is the only explanation for the loss of CDrB from T cells is not directly supported by the observations, as other efflux systems might be likely involved.
Legends of the figures are too synthetic and need to be much more informative about the data or concepts displayed. This is most critical for Figures S1 and S6. In supplementary figs. the word Figurer is a typo.
All the flow cytometric and cell sorting procedures should be described better, including the settings of the fluorescence detectors and the quality control procedures for excluding dead celles and cell-aggregates from the analysis and sorting. Indeed, the minor percentages of highly positive cells sorted sequentiaaly might include cell doublets, for instance. Also, the term FACS is not a synonimous of cell sorting, and should not be used, as it is the acronym for a commercial brand of flow cytometers and sorters.
Comments on the Quality of English LanguageMinor editing of English language is required
Author Response
Reviewer: 1
The manuscript by Cho et al reports the use of a novel fluorescent probe that is transported by two different transporters of the SLC family and allows the flow cytometric discrimination between murine T- and B-cells. This is a repliminary work with potential interest for the use of live-cell probes for cytometric studies. However, the manuscript has important limitations.
Answer: We appreciate with the reviewer’s positive and helpful comments. A detailed response is listed below.
The title is misleading and should be written more carefully. Actually, CDrB is not a B-cell selective probe, as it is taken up by SLCO1B3 which is present in both T- and B-cells. What makes the “labeling” selective of B cells is that T cells pump CDrB out due to overexpression of SLC25A41 mitochondrial transporter. Title should be more along the lines of selective identification of B cells based on the transport of CDrB probe.
Answer: We agree with the reviewer’s comments. By reflecting reviewer’s comments, the title was changed into “Novel fluorescent strategy to discriminate T and B lymphocytes via transporting system”.
The assumption that an alternative location (plasma membrane?) of SLC25A41 is the only explanation for the loss of CDrB from T cells is not directly supported by the observations, as other efflux systems might be likely involved.
Answer: We appreciate the constructive comments provided by the reviewers. In this article, we observed that SLC25A41-cloned cells exhibited a weak signal of CDrB. Additionally, T cells, which expressed SLC25A41 at high levels, displayed a dim intensity of CDrB, despite also expressing SLCO1B3, a known influx transporter. Based on these observations, we hypothesized that SLC25A41 may actively extrude its substrate, thereby suggesting a reversible localization. We believe that this finding contributes to exploring previously uninvestigated aspects of SLC25A41, which have not been well studied thus far.
Legends of the figures are too synthetic and need to be much more informative about the data or concepts displayed. This is most critical for Figures S1 and S6. In supplementary figs. the word Figurer is a typo.
Answer: We appreciate with helpful comments. We elaborated legends of the figures, and corrected a typo in supplementary figures.
All the flow cytometric and cell sorting procedures should be described better, including the settings of the fluorescence detectors and the quality control procedures for excluding dead celles and cell-aggregates from the analysis and sorting. Indeed, the minor percentages of highly positive cells sorted sequentiaaly might include cell doublets, for instance. Also, the term FACS is not a synonimous of cell sorting, and should not be used, as it is the acronym for a commercial brand of flow cytometers and sorters.
Answer: We agree with reviewer’s comments. We elaborated methods of flow cytometry and cell sorting procedures including fluorescence channels in more detail, and also described how we set gating strategies to carry out further experiments.

Reviewer 2 Report
Comments and Suggestions for Authors
The article "Novel fluorescent strategy of B cell selective probe" presents the development of a novel B cell selective probe with unique transport mechanism. The dye BODIPI was obtained and characterized using modern analytical methods. The authors, through SLC-CRISPRa systematic screening, identified two transporters, SLCO1B3 and SLC25A41. It was found that SLCO1B3 promotes probe transport into cells, while SLC25A41 actively exports the CDrB probe. This discovery can be utilized for selectively sensing lymphocytes using CDrB. While the research is of high quality, there are areas that need improvement, as detailed below:
1. References to other work should be included in the introduction to strengthen the research background. Nine references in the introduction are not sufficient to assess the current state of the field.
2. What is the quantum yield of the probe? In which solvent were the measurements made?
3. In the CDrB Synthesis section, add a scheme for obtaining the compound. Additionally, number the protons in the scheme for assignment in the NMR spectrum.
4. Lines 212-213: How were compounds selected for screening?
5. Line 269: Possibly conclusions.
6. The 1H NMR spectrum in Figure S7 is difficult to read in the 4.5 - 3.5 ppm region
Author Response
Reviewer: 2
The article "Novel fluorescent strategy of B cell selective probe" presents the development of a novel B cell selective probe with unique transport mechanism. The dye BODIPI was obtained and characterized using modern analytical methods. The authors, through SLC-CRISPRa systematic screening, identified two transporters, SLCO1B3 and SLC25A41. It was found that SLCO1B3 promotes probe transport into cells, while SLC25A41 actively exports the CDrB probe. This discovery can be utilized for selectively sensing lymphocytes using CDrB. While the research is of high quality, there are areas that need improvement, as detailed below:
Answer: We appreciate with the reviewer’s positive and helpful comments. A detailed response is listed below.
- References to other work should be included in the introduction to strengthen the research background. Nine references in the introduction are not sufficient to assess the current state of the field.
Answer: We agree with the reviewer’ comments. We put more references to support and strengthen the research background.
- What is the quantum yield of the probe? In which solvent were the measurements made?
Answer: We measured quantum yield of the probe (0.28) in methanol, and the detailed characters of the fluorophore was displayed in Figure S10.
- In the CDrB Synthesis section, add a scheme for obtaining the compound. Additionally, number the protons in the scheme for assignment in the NMR spectrum.
Answer: We added a synthesis scheme of CDrB, and assigned the number of all the protons in the synthesis section.
- Lines 212-213: How were compounds selected for screening?
Answer: We selected fluorescent-carbohydrate library for the screening, including 80 compounds, hypothesizing that main energy sources of carbohydrates may have distinctive selectivity along with fluorophore effects in the complex system. After the completion of screening analysis, we found out that CDrB showed a higher stain index among 80 compounds, implying a distant separation between two different groups. These contents were described and highlighted in line 229-235.
- Line 269: Possibly conclusions.
Answer: We agree with author’s comments. We rearranged those contents to a conclusion part, and put a new version in the discussion section.
- The 1H NMR spectrum in Figure S7 is difficult to read in the 4.5 - 3.5 ppm region
Answer: We agree with author’s comments. We enlarged and displayed 4.5-3.5 ppm region in Figure S7.

Round 2
Reviewer 1 Report
Comments and Suggestions for Authors
Authors have improved adequately the manuscript following my suggestions.
Comments on the Quality of English LanguageEnglish is acceptable
Reviewer 2 Report
Comments and Suggestions for Authors
Dear Authors,
I am satisfied with your response to my comments.